# Clinical Sign-Based Rapid Response Team Call Criteria for Identifying Patients Requiring Intensive Care Management in Japan

**DOI:** 10.3390/medicina57111194

**Published:** 2021-11-02

**Authors:** Reiko Okawa, Tomoe Yokono, Yu Koyama, Mieko Uchiyama, Naoko Oono

**Affiliations:** 1Emergency and Critical Care Center, Nagaoka Red Cross Hospital, Nagaoka 940-2085, Japan; re_ooka@ybb.ne.jp; 2Department of Nursing, Niigata University Graduate School of Health Sciences, Niigata 951-8518, Japan; yukmy@clg.niigata-u.ac.jp (Y.K.); uchiyama@clg.niigata-u.ac.jp (M.U.); 3Niigata College of Medical Technolgy, Niigata 950-2076, Japan; oono@niigata-coll-mt.ac.jp

**Keywords:** rapid response system, rapid response team, respiratory rates, inspired oxygen

## Abstract

*Background and Objectives*: For effective function of the rapid response system (RRS), prompt identification of patients at a high risk of cardiac arrest and RRS activation without hesitation are important. This study aimed to identify clinical factors that increase the risk of intensive care unit (ICU) transfer and cardiac arrest to identify patients who are likely to develop serious conditions requiring ICU management and appropriate RRS activation in Japan. *Materials and Methods*: We performed a single-center, case control study among patients requiring a rapid response team (RRT) call from 2017 to 2020. We extracted the demographic data, vital parameters, blood oxygen saturation (SpO_2_) and the fraction of inspired oxygen (FiO_2_) from the medical records at the time of RRT call. The patients were divided into two groups to identify clinical signs that correlated with the progression of clinical deterioration. Patient characteristics in the two groups were compared using statistical tests based on the distribution. Receiver operating characteristic (ROC) curve analysis was used to identify the appropriate cut-off values of vital parameters or FiO_2_ that showed a significant difference between-group. Multivariate logistic regression analysis was used to identify patient factors that were predictive of RRS necessity. *Results*: We analyzed the data of 65 patients who met our hospital’s RRT call criteria. Among the clinical signs in RRT call criteria, respiratory rate (RR) (*p* < 0.01) and the needed FiO_2_ were significantly increased (*p* < 0.01) in patients with severe disease course. ROC curve analysis revealed RR and needed FiO_2_ cut-off values of 25.5 breaths/min and 30%. The odds ratio for the progression of clinical deterioration was 40.5 times higher with the combination of RR ≥ 26 breaths/min and needed FiO_2_ ≥ 30%. *Conclusions*: The combined use of RR ≥ 26 breaths/min and needed FiO_2_ ≥ 30% might be valid for identifying patients requiring intensive care management.

## 1. Introduction

There has been a worldwide spread in the development and use of the rapid response system (RRS) based on the definition of clinical deterioration as movement “from one clinical state to a worse clinical state, which increases their individual risk of morbidity or death” [1]; moreover, deterioration events, such as cardiac arrests, frequently occur hours after abnormal vital signs [2]. The RRS, comprising a specialized rapid response team (RRT) or a medical emergency team, was developed in the 1990s to detect and prevent sudden lethal events in hospitalized patients with early intervention [3,4]. The RRS was developed for inpatients with various underlying diseases and at an increased risk of in-hospital cardiac arrest. There are several methods by which an RRT can be activated. These include single parameter systems and aggregate scoring systems where each vital sign parameter is assigned a score according to the level of derangement. These are then summed to generate an early warning score (EWS) with the response dictated by the value of the aggregate EWS. Other activation systems include patient, family or staff concern [5]. Many observational studies on RRSs have shown beneficial effects in terms of decreased hospital cardiac arrest and mortality rates [6]. However, a previous study reported that, the in-hospital mortality for RRT patients is approximately 25% overall but only 15% in patients without a limitation of medical therapy in the Australia [7].

Although anyone can call the RRS, the main requesters are nurses because the recognition and interpretation of physiological abnormality is primarily a nursing responsibility. Nurses are the healthcare professionals with the most direct patient contact and are responsible for both ongoing physiological assessment of patients and decisions regarding the frequency of patient assessment [8]. However, in some case, the RRS did not operate according to the required standards [9]. A delaying factor reported in RRT calls is that when there was awareness of the patients’ poor condition, the bedside staff included nurses felt that the clinical situation was under control in the ward setting. Another factor is fear of a negative or hostile reaction from colleagues for activating the RRS [10]. It is thought that the main reason why failure to activate the RRS is due to ambiguity of the justification for activating RRS. Therefore, previous studies showed that nurses’ perceptions of the call criteria influenced their use. Perceiving patient condition/call criteria as too sensitive [11] or non-specific [12,13] created alarm fatigue [14,15,16]. For effective function of the RRS, prompt identification of patients at a high risk of cardiac arrest and RRS activation without hesitation are important.

In Japan, beginning in 2008, the Japanese Society for Quality and Safety in Healthcare has taken the lead in setting “rapid response and resuscitation” as the goal of the Japanese Coalition for Patient Safety, and ever since it has taken up the challenge of “establishment of RRS for physical changes,” an increasing number of hospitals are introducing RRTs, whose role is to prevent such sudden changes. However, there are no reports that systematically clarify the current status of RRS (RRT) operations in Japan, such as their penetration rate and outline in hospitals. This study aimed to identify clinical factors that increase the risk of intensive care unit (ICU) transfer and cardiac arrest to identify patients who are likely to develop serious conditions requiring ICU management and appropriate RRS activation in Japan.

## 2. Materials and Methods

This was a single center retrospective cohort study that was approved by the Niigata University Ethics Committee (approval ID: 2019-0073, the date of approval: 6 August 2019). Consent was obtained from all patients before data were acquired from their medical charts. In this study, we used the opt-out method. The disclosure document for this clinical study was provided through the website of Nagaoka Red Cross Hospital. This study was conducted in accordance with the principles of the Declaration of Helsinki. We used the RRS of Nagaoka Red Cross Hospital, a 592-bed hospital including a critical care center with an ICU and emergency ward. We included all patients whose conditions necessitated an RRT call from April 2017 to March 2020. Then, some cases that do not meet the criteria for RRT such as consultations relating to whether they would feel sick during examination or treatment, were excluded from analysis.

### 2.1. RRT Call Criteria

RRT call criteria adopted at Nagaoka Red Cross Hospital are shown in Table 1. These criteria were based on the Quick Sequential Organ Failure Assessment (qSOFA) [17] score, which is used to identify inpatients with suspected sepsis in an emergency department or a general ward, who are predisposed to poor outcomes outside the ICU. The “worried” criterion was added to the RRT call criteria because early clinical deterioration awareness and intervention are required as patients with sepsis easily undergo clinical deterioration and have a high risk for in-hospital mortality.

### 2.2. Data Collection

We extracted the following demographic and clinical data from patient medical charts on ICU admission: admission department, comorbidities, age, sex, length of hospital stay, and purpose of hospitalization. We obtained the following vital parameters from the medical records at the time of RRT call: Glasgow coma scale (GCS) score (to evaluate the consciousness level), RR, blood pressure, heart rate, body temperature, and blood oxygen saturation (SpO_2_); the use of supplemental oxygen therapy was also derived from the medical records. Additionally, the fraction of inspired oxygen (FiO_2_) was obtained from oxygen administration device and oxygen dose information in the medical records. Additionally, we investigated the job type of RRT requesters, requested location, reasons for request, and patient prognosis. The data were summarized and expressed as mean ± standard deviation or median and interquartile range.

### 2.3. Identification of Relevant Clinical Signs for Predicting the Progression of Clinical Deterioration

Based on the clinical course after RRT consultation, patients were divided into two groups to identify clinical signs that correlated with the progression of clinical deterioration: Group M included patients who presented a mild clinical course and were continuously observed after RRT intervention without further intensive treatment and without ICU or emergency ward transfer; Group S included patients who presented a severe clinical course, were transferred to the ICU or emergency ward, and developed sudden cardiac arrest or death within 24 h after RRT intervention.

### 2.4. Statistical Analyses

Patient characteristics in the two groups were compared using statistical tests based on the distribution. Student’s *t*-test was used when the unpaired continuous parameters were normally distributed and the Mann–Whitney U test was used for non-normally distributed parameters. Further, Fisher’s exact test was used for unpaired categorical endpoints, particularly with small samples.

Receiver operating characteristic (ROC) curve analysis was used to identify the appropriate cut-off values of vital parameters or FiO_2_ that showed a significant difference between-group. Multivariate logistic regression analysis was used to identify patient factors that were predictive of RRS necessity. Statistical analyses were performed using SPSS statistics (version 26, IBM Corp., Armonk, NY, USA), and the significance probability of all tests was set at *p* < 0.05.

## 3. Results

### 3.1. Demographic Characteristics

Of 92 patients for whom the RRT was called during the study period, we excluded 27 who did not meet the RRT call criteria; hence, 65 patients were finally included (Figure 1). The patient characteristics are shown in Table 2. Among the 65 patients, 47 and 18 patients categorized into Groups M and S, respectively.

There were 27 (57.4%) and 10 (55.6%) men in Groups M and S, respectively; however, there was no significant difference in sex between the two groups. The median ages were 83 years and 73.5 years in Groups M and S patients, respectively, although there was no significant age difference between the two groups. Regarding the presence of comorbidities, in Group M, 25 (53.2%), 11 (23.4%), and 6 (12.8%) patients had gastrointestinal/hepatobiliary, orthopedic, and cardiovascular diseases, respectively; in Group S, only 7 patients (38.9%) had gastrointestinal/hepatobiliary diseases, and 2 patients each (11.1% each) had orthopedic, circulatory, and respiratory diseases.

Regarding the purpose of hospitalization, in Group M, 20 (42.6%), 14 (29.8%), and 12 (25.5%) patients were hospitalized for surgery, gastrointestinal endoscopy/drainage, and close examination/conservative observation, respectively. Group S included 6 (33.3%); 4 (22.2%); and 2 (11.1%) patients who were hospitalized for close examination, conservation observation, and follow up; post-surgical follow up and gastrointestinal endoscopy/drainage; and cancer treatment; respectively. The presence of comorbidities and the purpose of hospitalization were comparable between the two groups.

### 3.2. RRT Request Status

The majority of RRT requesters in Groups M and S were nurses (45 [95.7%] and 17 [94.4%], respectively), with no significant difference in requester’s occupation between the groups (*p* > 0.99). Most requests were received from the general ward (40 [85.1%] and 15 [83.3%] in Groups M and S, respectively), with no significant difference in location between the groups (*p* > 0.99) (Table 3).

Regarding the reasons for RRT activation, there were many reasons for different RRT calls, and thus multiple answers to the requests were allowed. The most common reason for RRT activation was the presence of respiratory abnormalities. In Group S, respiratory abnormalities occurred in 11 (61.1%) patients, of whom 7 patients had a decreased oxygen saturation (SpO_2_) and 4 patients had tachypnea; in Group M, respiratory abnormalities occurred in 24 (51.1%) patients, of whom 15 patients had a decreased SpO_2_ and 9 patients had tachypnea (Table 4).

Moreover, we examined how well the RRT call criteria were met in the hospital, since many patients had complex abnormalities, and multiple answers to the requests were allowed. In the present study, patients without vital sign data were excluded from the analysis. Tachypnea was the most frequent RRT call criterion in both Groups M and S patients (51.1% vs. 87.5%, respectively). Decreased consciousness level was the second most frequent RRT call criterion in both Groups M and S (38.6% vs. 68.8%, respectively). Hypotension was the third most frequent RRT call criterion in both Groups M and S (23.9% vs. 44.4%, respectively) (Table 5).

Regarding patient physiological data at the time of RRT intervention, the median GCS score was 15.0 (14.0–15.0), which was normal or matched the RRT call criteria. The median RR was 24.0 (19.0–31.0) breaths/min, which was 2 breaths/min higher than the RR in the RRT call criteria. The mean systolic blood pressure was 114.2 ± 7.2 mmHg, which was 14.2 mmHg higher than the value in the RRT call criteria. Comparing the parameters of Group S versus Group M patients, the consciousness level was significantly lower (14.0 vs. 15.0, *p* = 0.03), RR (30.0% vs. 22.0%, *p* < 0.01) and needed FiO_2_ (%) was significantly higher (40.0% vs. 28.0%, *p* < 0.01) (Table 6).

### 3.3. Identification of the Clinical Signs for Predicting the Progression of Clinical Deterioration

During the RRT intervention, two of the three parameters that showed significant differences between Groups M and S were related to the respiratory status, we focused on RR and FiO_2_.

ROC curve analysis was used to identify both the appropriate RR and needed FiO_2_ for predicting clinical deterioration; the cut-off value of RR was 25.5 breaths/min (sensitivity 0.75, specificity 0.71) with an area under the curve (AUC) of 0.77 (CI: 0.63–0.90, *p* < 0.01) (Figure 2), and the cut-off value of the needed FiO_2_ was 30% (sensitivity 0.82, specificity 0.64) with an AUC of 0.78 (CI: 0.64–0.92, *p* < 0.01) (Figure 3). Based on the results of ROC curve analysis, the cutoff value of RR was rounded off to 26 breaths/min, and the prediction of clinical deterioration at this cut-off was compared with that at 22 breaths/min that was used as a hospital standard criterion. The odds ratios of predicting the progression of clinical deterioration at 22-breaths/min and 26-breaths/min RR cut-offs were 6.70 (95% CI: 1.36–32.92, *p* = 0.02) and 7.39 (95% CI: 2.01–27.16, *p* < 0.01), respectively. The sensitivity of predicting the progression of clinical deterioration at the 22-breaths/min RR cut-off was 0.88, and decreased to 0.75 at the 26-breaths/min cut-off; however, the specificity at the 26-breaths/min RR cut-off was higher than that at the 22-breaths/min cut-off (0.71 vs. 0.49, respectively) (Table 7).

We used a combination of RR and needed FiO_2_ to examine a more reliable condition for predicting the progression of clinical deterioration. The odds ratio for predicting the progression of clinical deterioration was 40.5 times higher (95% CI: 3.93–417.43, *p* < 0.01) when a combination of RR ≥ 26 breaths/min and needed FiO_2_ ≥ 30% (Table 8) were used.

## 4. Discussion

The most common reason for RRS activation is ward staff concern [18], and the clinical judgment of the ward staff may improve the prediction of patient clinical deterioration [19]. Many RRSs use single- or multi-parameter vital sign abnormalities as indications for RRS activation [19]; hypotension, tachycardia or bradycardia, tachypnea, and reduced consciousness level are often included in multi-parameter vital sign monitoring [18]. Our hospital included these parameters in patient evaluation prior to RRS activation, and therefore, we believe that RRS activation in the present study was based on standard practice and was valid for analysis.

We compared clinical signs included in the RRT call criteria between Group M and S patients. We found that consciousness level (the median GCS score) at the time of RRT intervention was normal or matched that in the RRT call criteria. The causes of decreased consciousness level include primary lesions due to direct brain damage and secondary lesions due to circulatory disorders and hypoxia [20]. In this study, considering that most participants requesting RRS had decreased consciousness levels due to the secondary lesions and that deterioration of respiratory conditions and circulatory insufficiencies occurred, which decreased consciousness levels, it is presumed that the RRS was often requested before the decline of consciousness level.

On the other hand, the systolic blood pressure at the time of RRS intervention was better than that in the RRT call criteria. A case-control study on whether vital sign measurements in patients with intrinsic diseases could identify the risk of cardiac arrest revealed that ≥27 breaths/min in 72 h increases the risk of cardiac arrest [21]. Even when the pulse rate and blood pressure results did not predict cardiopulmonary arrest, it is likely that a respiratory illness could occur in the early stages of patient exacerbation, followed by altered circulation, leading to cardiac arrest. In this study, there were many requests related to tachypnea and decreased SpO_2_ during RRS intervention; therefore, it is possible that the RRT was called before hypotension.

Conversely, the mean RR in our study was 24.7 breaths/min, which deviated from that in the hospital RRT call criteria by approximately 2 breaths/min. This suggests that the RRS is not operated according to the current requirements, as in previous studies [8]. In clinical practice, either tachypnea is apparent or it is difficult to detect, unless RR is measured daily. Respiratory measurements were not included in the prediction instructions for each department at the Nagaoka Red Cross Hospital. Therefore, even if there is an abnormality in RR, SpO_2_ is maintained and the abnormality may be overlooked. In general wards, one of the causes of death due to sudden changes is early respiratory acidosis, i.e., increase in RR associated with increase in minute ventilation seen in congestive heart failure, sepsis, and pulmonary embolism. This indicates the need for early awareness of changes in tachypnea. However, regarding SpO_2_ changes, although its values are maintained due to the increase in minute ventilation, it is a compensatory change and a pseudo-stable condition [22]. Nurses need to be aware that overlooking this symptom increases the risk of a serious condition.

Notably, three parameters (consciousness levels, RR, and FiO_2_) showed significant differences between Group M and S patients and could be used to predict the progression of clinical deterioration. Since two (RR and FiO_2_) of these parameters reflect the respiratory condition, we determined the cut-off values of these two parameters in predicting the progression of clinical deterioration. The cut-off values of RR and needed FiO_2_ for the aggravation of respiratory status were 25.5 breaths/min (rounded off to 26 breaths/min) and 30%, respectively. To our knowledge, our study is the first to reveal that FiO_2_, which was not included in the RRT call criteria, was related to the progression of clinical deterioration. Several studies have shown an association between delayed RRS activation and increased in-hospital mortality rates, length of hospital stay, and number of cardiorespiratory arrests, as well as high ICU admission rates [23,24,25,26,27]. Prompt RRS activation is warranted in patients who require RRT intervention. However, staff hesitation to make an RRS request is a major cause of delayed RRS activation [10,23], and therefore, it is necessary to clarify the RRT call criteria using validated clinical signs for effective function of the RRS. Based on our findings, consciousness level, RR, and FiO_2_ required may be the criteria that can effectively be used to identify patients who will progress to serious conditions and require ICU management and appropriate RRS activation.

For detecting the progression of clinical deterioration, the sensitivity was 73.3% when using criteria of RR ≥ 22 breaths/min and FiO_2_ ≥ 30%, and 60% when using criteria of RR ≥ 26 breaths/min and FiO_2_ ≥ 30%, therefore, 2 out of 15 Group S patients (13.3%) may be miss-judged as Group M patients if using the criteria of RR ≥ 26 breaths/min and FiO_2_ ≥ 30%. On the other hand, the specificity was 86.1% when using RR ≥ 22 breaths/min and FiO_2_ ≥ 30%, and 90.7% when using criteria of RR ≥ 26 breaths/min and FiO_2_ ≥ 30%, and miss-judgement mild patients as severe could be improved in 2 of 43 patients (4.7%) if using the criteria of RR ≥ 26 breaths/min and FiO_2_ ≥ 30%. The accuracy of both criteria was 82.8%, which was equivalent. This time, considering the current situation of facilities where RRT calling has not progressed due to reasons such as hesitation, we would like to recommend the combined use of RR ≥ 26 breaths/min and needed FiO_2_ ≥ 30% as a criterion that should never hesitate RRT calling because it shows higher specificity and 40.5 times higher criteria for detecting progression of clinical deterioration.

The main limitations of the present study include the retrospective nature of the study, as well as the small number of patients who underwent RRS activation in a single institution in Japan, thereby limiting the generalizability of the study results. It is necessary to examine the criteria based on these results with other facilities, and increase the sample size in order to perform multivariate analysis using the level of consciousness as well as other parameters and potential confounding factors.

## 5. Conclusions

In order for ward staff to improve the prediction of patient clinical deterioration by timely clinical judgment in the wards and to effectively activate the RRS, it is necessary to set appropriate RRT call criteria based on clinical signs. In order to identify patients requiring ICU management, it might be necessary for ward staff to assess consciousness levels, RR, and needed FiO_2_, as well as a combination of RR ≥ 26 breaths/min and needed FiO_2_ ≥ 30%.

## Figures and Tables

**Figure 1 medicina-57-01194-f001:**
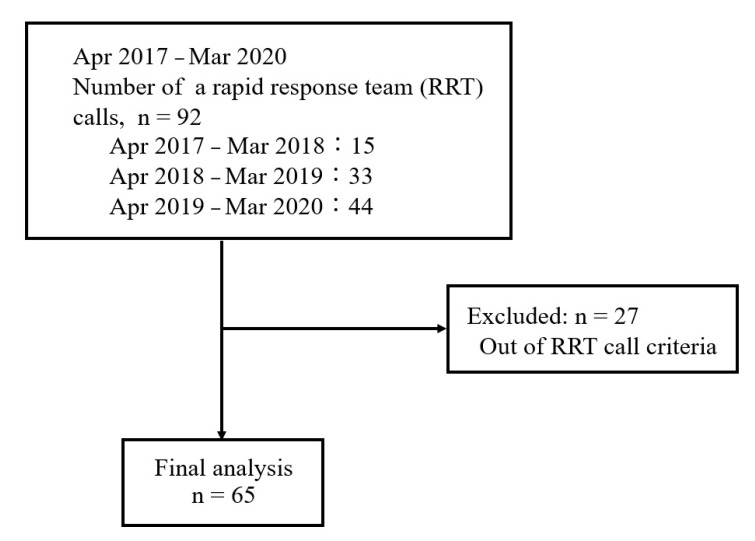
Flow chart of the study.

**Figure 2 medicina-57-01194-f002:**
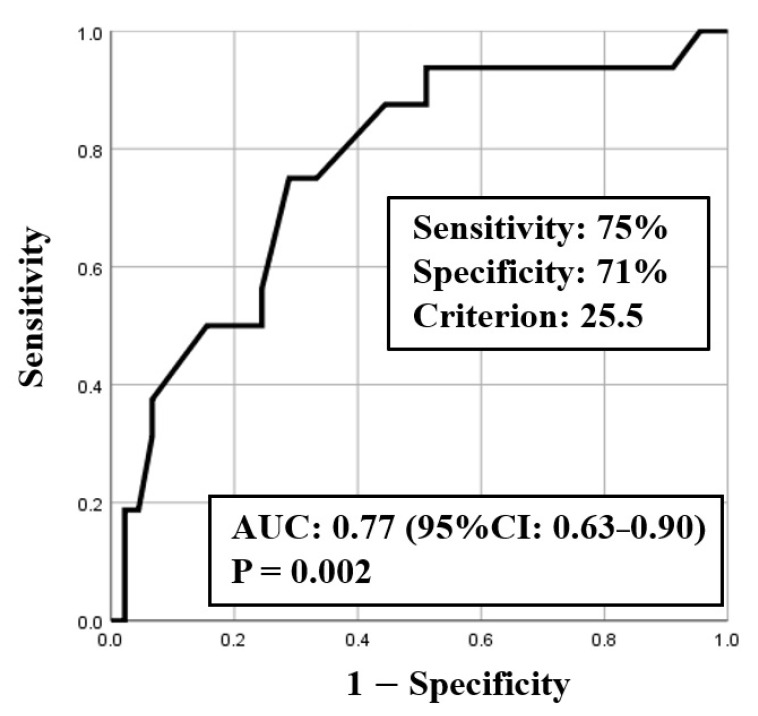
ROC curve for respiratory rate (RR).

**Figure 3 medicina-57-01194-f003:**
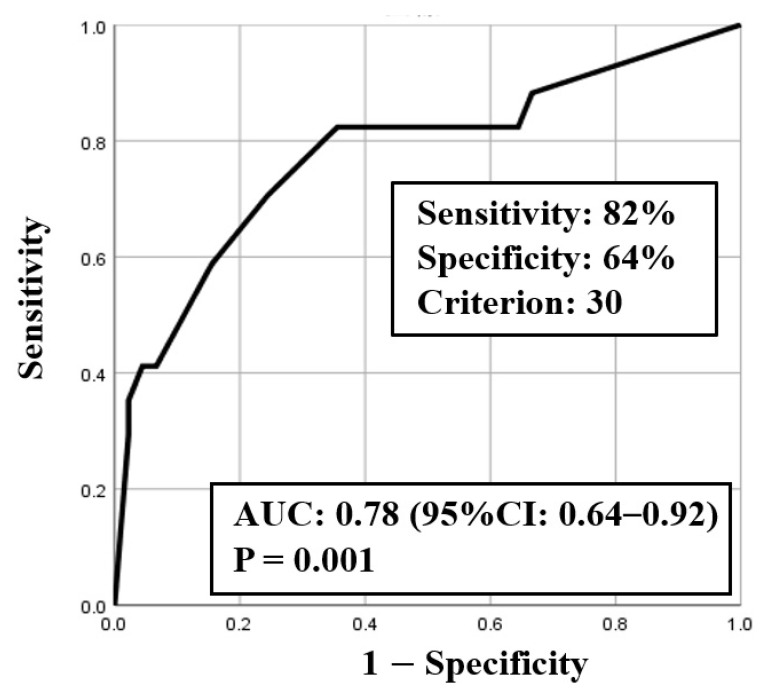
ROC curve for fraction of inspiratory oxygen (FiO_2_).

**Table 1 medicina-57-01194-t001:** RRT call criteria adopted at Nagaoka Red Cross Hospital.

RRT Call Criteria
(1) Glasgow coma scale (GCS) score ≤14 (decreased in consciousness level) (2) Respiratory rate (RR) ≥22 breaths/min (tachypnea) (3) Systolic blood pressure ≤100 mmHg (hypotension) (4) “Worried” criterion (nurses’ feelings relating to “something different, something strange”)

**Table 2 medicina-57-01194-t002:** Patient characteristics.

	Total (*n* = 65)	Group M * (*n* = 47)	Group S ** (*n* = 18)	*p*-Value
Sex				>0.99
Male	37 (56.9)	27 (57.4)	10 (55.6)	
Female	28 (43.1)	20 (42.6)	8 (44.4)	
Age (years)	78 (67–86)	83 (67–88)	73.5 (69–82)	0.28
Length of hospital stay (days)	5 (2–14)	5 (2–14)	5 (3–14)	0.63
Existing comorbidity				0.07
Gastrointestinal/hepatobiliary	32 (49.2)	25 (53.2)	7 (38.9)	
Orthopedic	13 (20.0)	11 (23.4)	2 (11.1)	
Cardiovascular	8 (12.3)	6 (12.8)	2 (11.1)	
Respiratory	5 (7.7)	3 (6.4)	2 (11.1)	
Others	7 (10.7)	2 (4.3)	5 (27.8)	
Purpose of hospitalization				0.11
Surgery	24 (36.9)	20 (42.6)	4 (22.2)	
Examination/observation	18 (27.7)	12 (25.5)	6 (33.3)	
Endoscopy/drainage	18 (27.7)	14 (29.8)	4 (22.2)	
Coronary IVR	2 (3.1)	1 (2.1)	1 (5.6)	
Dialysis	1 (1.5)	0 (0.0)	1 (5.6)	
Cancer treatment	2 (3.1)	0 (0.0)	2 (11.1)	

Data are given as the median (interquartile range) or as *n* (%). Categorical variables were compared using Fisher’s exact test, and continuous variables were compared using the Mann–Whitney U test. * Patients with a mild clinical course who continuously observed after rapid response team (RRT) intervention without further intensive treatment and without intensive care unit (ICU) or emergency ward transfer. ** Patients with a severe clinical course who were transferred to the ICU or emergency ward, and developed sudden cardiac arrest or death within 24 h after RRT intervention. IVR: interventional radiology.

**Table 3 medicina-57-01194-t003:** Background of rapid response team (RRT) request.

	Total (*n* = 65)	Group M * (*n* = 47)	Group S ** (*n* = 18)	*p*-Value
Requester’s occupation				>0.99
Nurse	62 (95.4)	45 (95.7)	17 (94.4)	
Physician	3 (4.6)	2 (4.3)	1 (5.6)	
Location				>0.99
General ward	55 (84.6)	40 (85.1)	15 (83.3)	
Examination/treatment room	10 (15.4)	7 (14.9)	3 (16.7)	

Data are given as *n* (%). Comparison between Group M and Group S was performed using Fisher’s exact test. * Patients with a mild clinical course who continuously observed after RRT intervention without further intensive treatment and without intensive care unit (ICU) or emergency ward transfer. ** Patients with a severe clinical course who were transferred to the ICU or emergency ward, and developed sudden cardiac arrest or death within 24 h after RRT intervention.

**Table 4 medicina-57-01194-t004:** Reasons for rapid response team (RRT) request.

	Total (*n* = 65)	Group M * (*n* = 47)	Group S ** (*n* = 18)
Decreased consciousness (GCS score ≤ 14 points)	10 (15.4)	5 (10.6)	5 (27.8)
Tachypnea (RR ≥ 22 breaths/min)	13 (20.0)	9 (19.2)	4 (22.2)
Hypotension (sBP ≤ 100 mmHg)	14 (21.5)	10 (21.3)	4 (22.2)
Worried criterion			
Decreased oxygen saturation	22 (33.8)	15 (31.9)	7 (38.9)
Abnormal body temperature	11 (16.9)	8 (17.0)	3 (16.7)
Abnormal heart rate	9 (13.8)	7 (14.9)	2 (11.1)
Acute pain	4 (6.2)	3 (6.4)	1 (5.6)
Chest Symptoms	3 (4.6)	3 (6.4)	0 (0.0)
Hypertension	1 (1.5)	0 (0.0)	1 (5.6)
Arrhythmia	1 (1.5)	1 (2.1)	0 (0.0)
Others	11 (16.9)	10 (21.3)	3 (16.7)

The numbers of reasons for RRT request were multiple answers to the requests. Data are presented as *n* (%). * Patients with a mild clinical course who continuously observed after RRT intervention without further intensive treatment and without intensive care unit (ICU) or emergency ward transfer. ** Patients with a severe clinical course who were transferred to the ICU or emergency ward, and developed sudden cardiac arrest or death within 24 h after RRT intervention. GCS: Glasgow coma scale; RR: respiratory rate; sBP: systolic blood pressure.

**Table 5 medicina-57-01194-t005:** Number of patients who matched the RRT request criteria.

	Total (*n* = 65)	Group M * (*n* = 47)	Group S ** (*n* = 18)
Decreased consciousness (GCS ≤ 14 points)	28 (43.1)/60 cases	17 (38.6)/44 cases	11 (68.8)/16 cases
Tachypnea (RR ≥ 22 breaths/min)	37 (60.7)/61 cases	23 (51.1)/45 cases	14 (87.5)/16 cases
Hypotension (sBP ≤ 100 mmHg)	19 (29.7)/64 cases	11 (23.9)/46 cases	8 (44.4)/18 cases

Patients without vital sign data were excluded from the analysis. Data are presented as *n* (%). * Patients with a mild clinical course who continuously observed after RRT intervention without further intensive treatment and without intensive care unit (ICU) or emergency ward transfer. ** Patients with a severe clinical course who were transferred to the ICU or emergency ward, and developed sudden cardiac arrest or death within 24 h after RRT intervention. GCS: Glasgow coma scale; RR: respiratory rate; sBP: systolic blood pressure.

**Table 6 medicina-57-01194-t006:** Physiological findings at rapid response team (RRT) request.

	Total (*n* = 65)		Group M * (*n* = 47)		Group S ** (*n* = 18)		*p*-Value
GCS (Points)	15.0 (14.0–15.0)	(*n* = 60)	15.0 (14.0–15.0)	(*n* = 44)	14.0 (12.0–15.0)	(*n* = 16)	0.03
RR (breaths/min)	24.0 (19.0–31.0)	(*n* = 61)	22.0 (18.0–28.1)	(*n* = 45)	30.0 (24.5–36.1)	(*n* = 16)	<0.01
sBP (mmHg)	114.2 ± 7.2	(*n* = 64)	116.4 ± 26.9	(*n* = 46)	108.6 ± 28.1	(*n* = 18)	0.31
HR (beats/min)	95.4 ± 24.1	(*n* = 64)	91.9 ± 23.6	(*n* = 46)	104.2 ± 24.0	(*n* = 18)	0.07
BT (°C)	37.2 ± 1.2	(*n* = 44)	37.4 ± 1.1	(*n* = 31)	36.9 ± 1.5	(*n* = 13)	0.20
SpO_2_ (%)	94.1 ± 4.6	(*n* = 62)	94.7 ± 4.0	(*n* = 45)	92.4 ± 5.6	(*n* = 17)	0.07
FiO_2_ (%)	28.0 (21.0–40.0)	(*n* = 62)	28.0 (21.0–35.1)	(*n* = 45)	40.0 (32.0–90.1)	(*n* = 17)	<0.01

Patients without vital signs were excluded from the study. Data are expressed as mean ± standard deviation or median (interquartile range). Comparisons between Groups M and S were performed using the Student’s *t*-test or the Mann–Whitney U test based on the distribution of the data. * Patients with a mild clinical course who continuously observed after RRT intervention without further intensive treatment and without intensive care unit (ICU) or emergency ward transfer. ** Patients with a severe clinical course who were transferred to the ICU or emergency ward, and developed sudden cardiac arrest or death within 24 h after RRT intervention. GCS: Glasgow coma scale; RR: respiratory rate; sBP: systolic blood pressure; HR: heart rate; BT: body temperature; SpO_2_: percutaneous oxygen saturation; FiO_2_: fraction of inspiratory oxygen.

**Table 7 medicina-57-01194-t007:** Cut-off value for respiratory rate and clinical deterioration.

	OR (95% CI)	*p*-Value	Sensitivity	Specificity
RR cut-off				
22 (breaths/min)	6.70 (1.36–32.92)	0.02	0.88	0.49
26 (breaths/min)	7.39 (2.01–27.16)	<0.01	0.75	0.71

OR: odds ratio; 95% CI: 95% confidence interval; RR: respiratory rate.

**Table 8 medicina-57-01194-t008:** The combination of respiratory rate and fraction of inspiratory oxygen for predicting clinical deterioration.

	OR (95% CI)	*p*-Value
RR ≥ 26 (breaths/min) and FiO_2_ < 30%	4.00 (0.32–50.23)	0.28
RR < 26 (breaths/min) and FiO_2_ ≥ 30%	4.50 (0.42–48.53)	0.21
RR ≥ 26 (breaths/min) and FiO_2_ ≥ 30%	40.50 (3.93–417.43)	<0.01

Patients without information of FiO_2_ were excluded from this table. OR: odds ratio; 95% CI: 95% confidence interval; RR: respiratory rate; FiO_2_: fraction of inspired oxygen.

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
