# Peer review of "Clinical Sign-Based Rapid Response Team Call Criteria for Identifying Patients Requiring Intensive Care Management in Japan"

_medicina, 2021, doi:10.3390/medicina57111194_

Round 1

Reviewer 1 Report

I read this work with great interest, the article deals with a very interesting topic and the results bring useful information for the correct identification of patients who may require intensive care management.
Although well structured I suggest minor revisions which in my opinion could bring greater clarity to the reader.
In particular, I suggest reporting the RRS and RRT criteria in a table.
Furthermore, it would be useful to specify what is meant by "worried criterion" already in the paragraph on methods.

Author Response

Dear reviewer1

Thank you for the thoughtful and constructive feedback you provided regarding our manuscript. I answered the following comments.

Point 1: I suggest reporting the RRS and RRT criteria in a table. Furthermore, it would be useful to specify what is meant by "worried criterion" already in the paragraph on methods.

Response 1: Thank you for your wonderful suggestion. We added the following table.

Table1. RRT call criteria adopted at Nagaoka Red Cross Hospital.

RRT call criteria

1) Glasgow coma scale (GCS) score ≤14 (decreased in consciousness level)

2) Respiratory rate (RR) ≥22 breaths/min (tachypnea)

3) Systolic blood pressure ≤100 mmHg (hypotension)

4)“Worried” criterion (nurses’ feelings relating to "something different, something strange," )

Reviewer 2 Report

This is a well-done study of rapid response patients in Japan. The authors used two groups of patients, ones with "Mild" and "Severe" conditions. The way the manuscript is written, the group S had to transfer to ICU or emergency ward ?AND? develop cardiac arrest or death within 24-hours. I suspect they meant OR developed cardiac arrest or death (line 117). 

The analysis is well done, however, as the authors mention the N is low (n=67), and as one can see the ROC curves are "choppy." Despite this fact, this is a nice study. My major criticism is the need for a confusion matrix (i.e. 2x2 table) that would classify the number and percent of patients that would be missed if they change the criteria from RR of 22 to RR of >=26 and FiO2 > 30%. 

Also the authors in the discussion mention that "prompt RRS activation is required, but they don't discuss any solutions or suggestions.  There are already defined criteria at Nagaoka Red Cross Hospital (line 88). If the physicians and nurses are not calling based on these criteria, why would call in the new criteria? If the authors want to suggest an automated intervention -- i.e. automatic paging, they should elaborate this in the discussion and include a literature review around these automated alerts.  More importantly, they don't know how they would apply to all the other patients of the hospital, since the analysis was limited to 67 patients. All these should be discussed and included in the limitations 

Also, please double-check your english. there are several spelling mistakes as below. 

  • The numbers appear very low for a 3-year period. For example, in our institution, we achieve those RRT numbers in one month.
  • Please clarify what you mean by "temporary" in line 311: Did you mean primary? 
  • Change "one of the causes of deaths"  line 333 to "one of the causes of death" 

Author Response

Dear reviewer 2

Thank you for the thoughtful and constructive feedback you provided regarding our manuscript. I answered the following comments.

Point 1: My major criticism is the need for a confusion matrix (i.e. 2x2 table) that would classify the number and percent of patients that would be missed if they change the criteria from RR of 22 to RR of >=26 and FiO2 > 30%..

Response 1: Thank you for your proper suggestions. According to your comments, we calculated sensitivity, specificity and accuracy of each criterion, and we added the description bellow into the Discussion, instead of adding table.

"For detecting the progression of clinical deterioration, the sensitivity was 73.3% when using criteria of RR ≥ 22 breaths/min & FiO₂ ≥ 30%, and 60% when using criteria of RR ≥ 26 breaths/min & FiO₂ ≥ 30%, therefore, 2 out of 15 Group S patients (13.3%) may be judged as Group M patients if using the criteria of RR ≥ 26 breaths/min & FiO₂ ≥ 30%. On the other hand, the specificity was 86.1% when using RR ≥ 22 breaths/min & FiO₂ ≥ 30%, and 90.7% when using criteria of RR ≥ 26 breaths/min & FiO₂ ≥ 30%, and miss-judgement mild patients as severe could be improved in 2 of 43 patients (4.7%) if using the criteria of RR ≥ 26 breaths/min & FiO₂ ≥ 30%. The accuracy of both criteria was 82.8%, which was equivalent. This time, considering the current situation of facilities where RRT calling has not progressed due to reasons such as hesitation, we would like to recommend the combined use of RR ≥ 26 breaths/min and needed FiO2 ≥ 30% as a criterion that should never hesitate RRT calling because it shows higher specificity and 40.5 times higher criteria for detecting progression of clinical deterioration. "

Point 2: Also the authors in the discussion mention that "prompt RRS activation is required, but they don't discuss any solutions or suggestions.  There are already defined criteria at Nagaoka Red Cross Hospital (line 88). If the physicians and nurses are not calling based on these criteria, why would call in the new criteria? If the authors want to suggest an automated intervention -- i.e. automatic paging, they should elaborate this in the discussion and include a literature review around these automated alerts. 

Response 2: Thank you for making important points. Although there are already defined criteria at Nagaoka Red Cross Hospital, as the reviewer has pointed out, however, perhaps ward nurses were hesitant to call RRT on this criterion, so this time we did this research on clearly showing conditions at higher risk of aggravation. The purpose was to be able to perform RRT calling without any hesitation.

Point 3: More importantly, they don't know how they would apply to all the other patients of the hospital, since the analysis was limited to 67 patients. All these should be discussed and included in the limitations

Response 3: We added the following underlined text to the last paragraph of the discussion.

"The main limitations of the present study include the retrospective nature of the study, as well as the small number of patients who underwent RRS activation in a single institution in Japan, thereby limiting the generalizability of the study results. It is necessary to examine the criteria based on these results with other facilities, and increase the sample size in order to perform multivariate analysis using the level of consciousness as well as other parameters and potential confounding factors. "

Point 4: The numbers appear very low for a 3-year period. For example, in our institution, we achieve those RRT numbers in one month.

Response 4: There is no mistake in the number in 3 years at Nagaoka Red Cross Hospital, and there may be few RRT callings in Japanese hospitals due to medical circumstances such as the attending physician and ward on-duty doctor directly responding.

Point 5: Please double-check your English. there are several spelling mistakes as below.

1) Please clarify what you mean by "temporary" in line 311: Did you mean primary?

    As you pointed out. I fixed "temporary" to "primary".

2) Change "one of the causes of deaths" line 333 to "one of the causes of death"

Response 5:

1) Thank you very much. According to your wonderful advice, we correct "temporary" in line 311 to "primary".

2) Thank you for pointing out our mistake. According to your comment, we correct "one of the causes of deaths" in line 333 to "one of the causes of death".

Round 2

Reviewer 2 Report

The authors have addressed my points. Recommend acceptance. 

Author Response

Dear. Reviewer 2 :

Thank you for your helpful and appropriate comment.

According to your comment, we have corrected the description of line 256-258 as "The odds ratios of predicting the progression of clinical deterioration at 22-breaths/min and 26-breaths/min RR cut-offs were 6.70(95% CI: 1.36-32.92, P=0.02) and 7.39(95% CI: 2.01-27.16, P<0.01), respectively".

Sincerely, 

Tomoe Yokono
